# Challenges to nutrition management among patients using antiretroviral therapy in primary health 'centres' in Addis Ababa, Ethiopia: A phenomenological study

**Helen Ali Ewune** [ORCID]*, **Kassa Daka, Befekadu Bekele, Mengistu Meskele**

School of Public Health, Wolaita Sodo University, Wolaita Sodo, Ethiopia

* bravohelu@gmail.com

## Abstract

### Introduction

Nutritional management is a fundamental practice of concern to all patients infected with the human immunodeficiency virus (HIV). The nature of HIV/AIDS and malnutrition impacts are interlocked and intensify one another.

### Objective

This study aimed to explore nutrition management challenges among people living with HIV on antiretroviral therapy (ART) in primary health centres in Addis Ababa, Ethiopia.

### Methods and materials

We used a hermeneutic (interpretive) phenomenological study design. The study used in-depth interviews to describe lived experiences among adult patients aged 18 and above. We selected the participants purposively until the saturation of the idea reached. We maintained the scientific rigor and trustworthiness by applying credibility, transferability, dependability, and conformability, followed by translation and re-reading of the data has been achieved. The data have been analyzed through inductive thematic analysis assisted by NVIVO version 12 pro software.

### Result

Nutrition management challenges for HIV patients have been described using six significant themes. The major themes were: acceptance of the disease and the health status; facilitators and barriers to treatment adherence; behavioural changes in eating patterns; experience of food insecurity issues; nutrition knowledge; and support. The themes have explained how patients using ART have been challenged to manage their nutrition ever since their diagnosis. Of all challenges, food insecurity is found to be the core reason for poor nutrition management.

**Data Availability Statement:** All relevant data are within the manuscript and its Supporting Information files.

**Funding:** Norwegian agency of development (NORAD) and Wolaita Sodo University College of Health Science have funded this research as part of Master's Degree research.

**Competing interests:** The authors have no competing interests.

## Conclusion and recommendation

We found that many factors in managing their nutrition challenged patients with HIV. There should be an increasing interest in managing food insecurity issues as food insecurity has been strongly related to other factors.

## Background

Nutritional management is a fundamental concern for all patients infected with the human immunodeficiency virus (HIV) [1].

Globally, over 800 million individuals are chronically suffering from malnutrition, specifically under nutrition, and more than 35 million individuals are living with human immunodeficiency infection (HIV) [2].

Energy requirements in HIV patients increase by 10% to keep up body weight and physical movement in asymptomatic HIV-infected people. During symptomatic HIV, and in this manner during AIDS, energy prerequisites increase by approximately 20% to 30% to maintain body weight [3].

Nutrition Management covers issues of food insecurity and malnutrition, which poses challenges to antiretroviral treatment (ART) adherence in resource-poor settings and is often cited by PLWHA [4].

People living with HIV/AIDS lifestyle find it hard to manage their nutrition, but knowledge could help construct principles for promoting nutritional intervention. Adequate nutritional support is crucial to ensuring optimal ART uptake, adherence, and outcomes [5].

HIV infection is worldwide; however, most cases are commonly registered in low- and middle -income countries, predominantly in Sub-Saharan Africa (SSA). Also in Ethiopia, 1.5% of young individuals aged 15–49 are infected with HIV [6].

HIV influences the well-being and prestige of a person. Moreover, the socio-economic well-being of individuals infected with HIV is adversely affected, and it also affects the economic growth of a country which in turn affects the development of a nation [7, 8].

Ethiopia is one of the countries badly hit by HIV infection and malnutrition. Although trained counsellors should give nutrition intervention, a study conducted in Northwest Ethiopia found that dietary counselling was not practised [9, 10].

The seriousness of malnutrition among PLWHA on ART is continually escalating. Studying the experience of patients using ART will help patients to strengthen their healthy lifestyle. Therefore, this study aimed to explore challenges to nutrition management among patients on ART in primary health 'centers' in Addis Ababa, Ethiopia.

## Methods and materials

### Study area and setting

The current study was conducted in Addis Ababa, the capital city of Ethiopia. It is also the largest city in the country by population, with a total population of 3,384,569 according to the 2007 census. There are currently nine regional states and two chartered cities, Addis Ababa, and Dire Dawa. Among the nine regions, HIV prevalence is highest in Gambella (4.8%) followed by Addis Ababa city administration, the second-highest city from the overall country with HIV prevalence (3.4%), Dire Dawa (2.5%), and Harari (2.4%).

## Research design

We used an Interpretative Phenomenological Analysis (IPA) design. This method primarily works with transcripts of semi-structured interviews. The analyzed themes were converted into a narrative account and also supported with verbatim extracts from participants.

## Inclusion criteria

Patients who volunteered to participate in this study, were at least 18 years old or above at the time of diagnosis (we hoped to minimize recall bias). Participants for the current study permanently resided in Addis Ababa. They were aware of their HIV status and willing to consent and speak about their experience. Participants had at least prior exposure to severe weight loss secondary to malnutrition and recovered with supplemental foods provided at the health centre. They had lived at least one year with HIV with a full year of ten years.

## Exclusion criteria

Exclusion criteria included patients with severe illness.

## Data collection procedure

Open-ended semi-structured interview guides were provided before the data collection; the data collector (the researcher) piloted the instruments with five patients using ART. Adjustments were made for flow, content, terms used, prompts, and instructions. We translated the tools into the local language, Amharic.

An interview meeting was scheduled at a time and place suitable for the participant. Probes helped to follow a route of an investigation initiated by the participant. The primary author conducted the interview. We selected participants purposively until saturation of the data was reached.

Audio recordings and notes were made. There were two audio tape recorders in case of equipment failure. After all the questions were addressed, the researcher asked the participants if there was anything additional they wanted to discuss. The body language, verbal, and non-verbal cues were recorded using notes immediately after the interview. The researcher and other transcribers performed transcription. After the interview, transcription, and initial data analysis, the participant was contacted for transcription verification. The participant needed to verify that the transcription was exactly what she/he had said and verify the participant's description of her lived experience. A total of 28 in-depth interviews were conducted using the most appropriate technique for data collection.

## Ethical considerations

Ethical approval and clearance were obtained from the Institutional Review Board(IRB). Formal permission was obtained from the Addis Ababa Health Office for the respective health centres where the study was conducted.

The participants were reassured that none of the real identities would be revealed. All questions and concerns were addressed before signing informed consent. Participants were informed that there was no direct benefit from this study except an opportunity to talk freely about their condition, which could be either a distressing or helpful experience. Also, they were informed about the expected duration of the interview.

The location ensured the participant's privacy and was mutually agreed. The participants were asked if they had any questions after completing the interview question guides. Written informed consent was received from the participants. The interview participants had the right

to withdraw from the study at any time. If they did withdraw, there was no penalty, and the service they were receiving was not affected. They may refuse to answer any question or have the recording device stopped. If they did, the information would still be used unless they withdrew completely.

## Data analysis

The recorded interviews were transcribed in Amharic and translated into English. The data were imported to NVIVO version 12 pro software following coding and synthesis. The current study summarized a large amount of data with the essential features of the interviews. As per Cress well's description of six significant steps in analyzing data in phenomenology [11], the current study has followed six significant analysis steps. First, the data were managed by creating and organizing files so that they could be accessed easily for analysis. Second, we read and re-read the transcriptions, making notes in the margins identifying any sub-themes and emerging ideas to form baseline codes. Third, we described personal experiences throughout the period, attempting to identify the core meaning or essence of the phenomena, the concepts, or themes derived from the data. Fourth the data were classified by developing relevant statements. The statements were grouped according to significance and meaning in the coding process. Step five was the interpreting phase, which is a written description to answer what happened and how the phenomenon was experienced. Thus, the essence of the experience of the 'participants' was revealed. The final step was summarizing the study's findings, and this has been also been supported with additional paragraphs that intend to interpret the meaning of the experience.

## Scientific rigor and quality assurance

**Credibility.**   In the current study, this was achieved by prolonged engagement, peer debriefing, and member check. It has been secured in collecting data, observing, and interviewing to gain in-depth knowledge.

**Dependability.**   The study's reliability in the data collection and data analysis process was assured by accurate documentation, avoiding spelling and grammatical errors that will reduce the quality of work, and a detailed description of producers is included.

**Conformability.**   The data were checked by independent people based on the data accuracy, relevance and meaning.

**Transferability.**   Transferability was ensured by providing a detailed description of the study setting, thereby providing other readers opportunities to contextualize to another area.

## Operational definitions

**The challenge of nutrition management is not being able to access and eat foods that help maintain the optimum level of nutritional status of HIV patients on ART, which can help them** perform regular activities.

**Nutrition management** is a phenomenon of accessing and eating foods that help maintain the optimum level of nutritional status of HIV patients on ART, which can help them perform regular activities.

**Weight loss**: is referred to in this study as a reduction of weight among patients with HIV on ART secondary to malnutrition when other weight loss factors are ruled out.

**PLHIV**: Refers to the people living with HIV on ART 18 and above years of age and diagnosed as a case of HIV as per the records and who are attending selected ART Centers for treatment in primary health centres.

## Result

### Participant characteristics

The study included 28 HIV patients on ART who lived in Addis Ababa. Twenty-eight participants were recruited from three governmental health centers. The 'participants' age varied from 29 to 61, with a mean age of 42 years. The participants reported that they were living with this disease from two to ten years. However, all the participants were living in a different Sub-city of Addis Ababa. Twenty-one of the women stated that they were Orthodox Christian by religion; six participants were Protestant Christian and only one was Muslim. Their marital status was reflected in four states: 11 were married, five were divorced, four were single, and eight were widowed. Regarding their employment status, eight of the 28 participants were unemployed, two were governmentally employed; and 17 were employed in the private sector as drivers, security staff, waitress, and daily labourer. One was a commercial sex worker (Table 1).

### Themes

Six major themes were identified from 'participants' testaments: acceptance of disease and health status; facilitators and barriers of drug adherence; behavioral change in eating pattern;

**Table 1. Socio-demographic characteristics of research participants.**

| Participant code | Sex | Age | Religion | Employment Status | Marital status | Number of years with HIV on ART |
|---|---|---|---|---|---|---|
| K1 | F | 56 | OC | Commercial sex worker | Single | 9 years |
| Y2 | F | 45 | PC | Daily laborer | Divorced | 8 years |
| A3 | F | 50 | PC | Unemployed | Married | 8 years |
| K4 | F | 35 | OC | Self Employed | Married | 8 years |
| Y5 | F | 30 | OC | Self Employed | Widowed | 8 years |
| A6 | F | 55 | OC | Unemployed | Divorced | 4 years |
| A7 | F | 40 | OC | Unemployed | Married | 10 years |
| K8 | F | 38 | OC | Unemployed | Single | 4 years |
| A9 | F | 29 | OC | Unemployed | Singe | 10 years |
| Y10 | F | 36 | OC | Waitress at Private | Divorced | 6 years |
| K11 | F | 30 | PC | Waitress at Private | Widowed | 3 years |
| K12 | M | 41 | OC | Daily laborer | Married | 5 years |
| Y13 | M | 42 | OC | Daily laborer | Widowed | 4 years |
| K14 | M | 45 | OC | Daily laborer | Widowed | 3 years |
| Y15 | M | 40 | OC | Driver at Private | Married | 3 years |
| Y16 | M | 55 | M | Driver at Private | Married | 4 years |
| K17 | M | 41 | PC | Driver at Private | Married | 6 years |
| K18 | M | 39 | OC | Driver at Private | Single | 6 years |
| A19 | M | 40 | OC | Driver at Private | Widowed | 2 years |
| K20 | M | 33 | OC | Driver at Private | Widowed | 8 years |
| K21 | M | 36 | PC | Government Employee | Widowed | 7 years |
| Y22 | M | 37 | OC | Government Employee | Married | 10 years |
| K23 | M | 46 | OC | Security at Private | Married | 2 Years and 8 months |
| Y24 | M | 61 | OC | Unemployed | Divorced | 6 years |
| A25 | M | 50 | OC | Security at Private | Married | 8 years |
| K26 | M | 44 | OC | Self Employed | Married | 8 years |
| Y27 | M | 38 | OC | Unemployed | Divorced | 6 years |
| A28 | M | 39 | PC | Unemployed | Widowed | 7 Years |

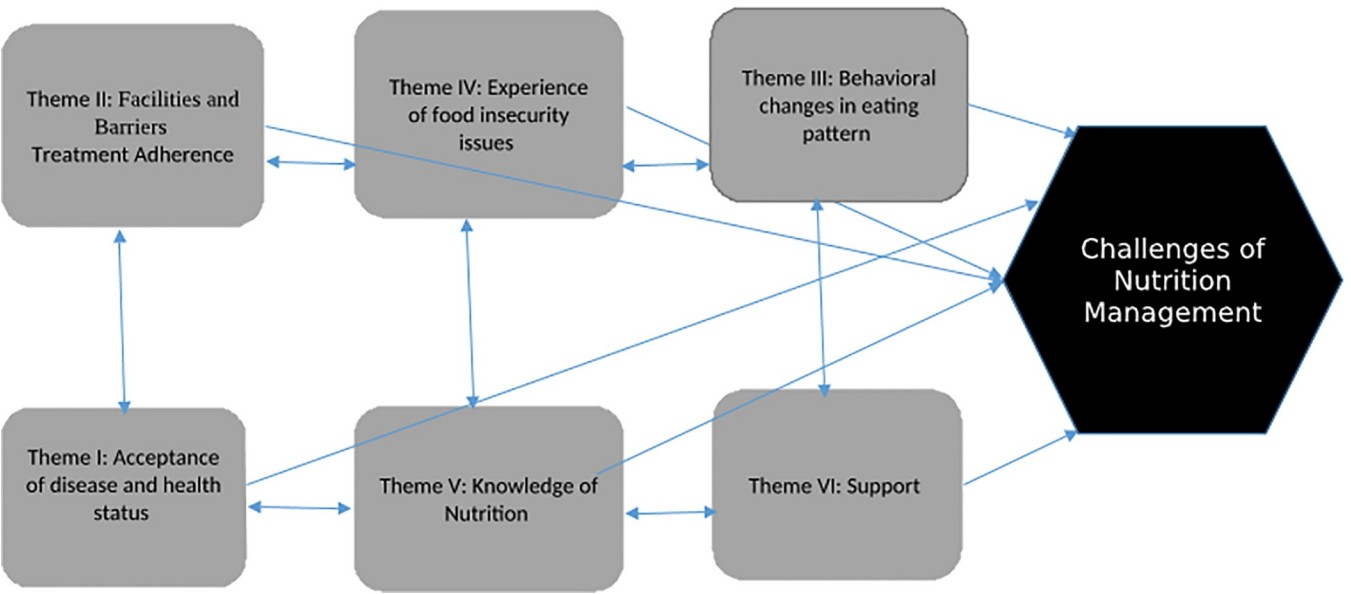

**Fig 1. Thematic map of identified themes: Interaction between themes and burden to manage nutrition among HIV patients on ART.**

the experience of food insecurity issues; nutrition knowledge; and support were assembled as a major challenge in their life course ever since their diagnosis. Also, it shows what their challenges to control their healthy lifestyle were. These themes were not restricted to a specific participant because the experiences of a single participant could belong to more than one theme (Fig 1)

**Theme I: Acceptance of disease and health status.** This theme is concerned about how accepting a new phenomenon was a challenge to having better nutrition management. Participants said accepting the positive status was a challenge to having positive behaviour towards nutrition. The phenomenon was beyond the support and education given by health professionals at the health centre in this regard. As a result, patients stopped taking any medication, which led them to have health and physical impairment. The other factor which they mentioned was spirituality. For some, spirituality was mentioned as a restricting factor as they spent their time in the holy water without taking any medications. For others, their spirituality had helped them accept their health status. Besides, participants stated their test (virus detection) seeking behavior was poor, resulting in a denial of their positive results so that they 'couldn't manage their nutritional status.

*"How could I start taking medication and eating foods if I can't accept my positive result . . .. for me. . .it was a terrible situation, and I was born again when I start accepting the reality. . .I have started taking my foods and medications after that" Female Unemployed, Aged 55*

*". . .I went to holy water, which is 30km far from Addis Ababa. When I was entered to the monastery, I have just stopped all foods for the blessings. I took food. . .well it was "Kita" after one day I didn't take any medication at that time. I thought I could be cured, and I suffered for 15 days" Male self-employed Aged 44*

**Theme II: Facilitators and barriers of treatment adherence.** This theme represents the anticipatory beliefs about how food, drug, and patient's drug adherence have challenged

patients. Participants described facilitators and barriers to their nutrition management as their experience was interlinked one to the other to eat well. They said whenever they are not taking the drugs; they had difficulty eating their meals regularly. A significant influence on their nutrition management was treatment know-how and passive tolerance of the treatment's side effects.

*"Taking a medication appropriately is meant a lot for me. . .what I want to tell you about my drug is how it helped me to live healthily. I need to live healthily and work my regular activity. Nevertheless, I have started it lately; I know how I benefited and survived my health condition. Look. . .the drug is also a food that enables us to eat regular food" Male Government employee Aged 37*

Participants did not know how ART drugs work actively; as a result, they were reluctant to adhere to the treatment. Moreover, 'participants' also stated that the drug had direct and numerous effects on their daily eating pattern.

*"I started eating and gaining weight after I have started the drug. I sleep well at night. I went out for work out, and I earn at least a little and that is because I started to work" Male Driver "Previously, when I started the medication it was 3 in kinds, and that has to be taken every three hours . . .though it was very challenging, but I have accepted it. I got no choice. I have to take the drug, and when it is challenging for me, what can I do? I just simply accept it" Male Unemployed Aged 38*

Participants' commonly perceived a strong relationship between their situation and how it affected their nutrition management. They were compliant with their prescribed medications. Other participants voiced, particularly those with other comorbid diseases, and described taking medication inconsistently. For most of the participants, it seemed that almost all became tolerant to ART treatment's side effects from time to time even though it was a barrier for them to achieve a healthy lifestyle.

*"Nevertheless, I have started it lately; I know how I benefited and survived my health condition. Look. . .It is also a food which enables us to eat the other side of food (regular meal)" Male Government employee Aged 36*

**Theme III: Behavioural change in eating pattern.** Behavioural change in eating patterns is distinct from defining experiences, such that eating habits in this theme capture the on-going and dynamic process affecting the quality of life of participants in the current study. It consists of factors influencing their nutrition management as they had changed their eating behaviour. Of the elements, loss of cultural eating practice because of eating pattern changes and the use of supplemental foods was explored. Participants expressed eating pattern changes and how that was a significant challenge for them to control their nutrition. Participants had similar experiences with eating pattern.

*". . .Of course, there was a change in my eating ever since my diagnosis. Yes. . .that was one of the stressing events so that one might not be able to think to manage his consumption. . .I knew it. . . they changed their eating, and I did. I ate food intentionally considering my status. . .HIV. . . but it was challenging because of my little understanding "Female unemployed Aged 55*

"*You have to take care of yourself, but you are in the meantime going apart from your social food ceremonies. . .I ate something before three years in   Meskel' festival, but I was sick and taken to a health centre the very next day*" *Female governmental employee Aged 50*

Provision of nutritional supplements was one of the supports given at the health center. Regarding the supplements test, participants explained difficulties of adapting it and refused to take it. for patients who had been given the supplementary food.

"*Taking that . . ...what was that . . ..there is food which is packed, the test is similar to a nut. It is challenging to eat for some of us because It has a different test. . .but it was not easy to take though. . . It disgusts me yak. . ."* *Female Daily laborer Aged 45*

**Theme IV: Experience of food insecurity issues.**    Participants raised two significant aspects of their food insecurity experience: their purchasing power related to what is available in a nearby market and the need to travel to buy cheaper foods and groceries outside their compound. It is also reported as a barrier to adhering to their drugs. Participants said that the limited availability of food affected their lifestyle change. On the contrary, some participants also said there was an availability of food, but their major problem was they could not afford it.

"*It is difficult sometimes to talk about food when there is nothing you can do about it. How could I help to manage my nutrition when I got nothing to eat. You might not be able to find anything which you can afford to eat. Tell me how I can take my medication. But health professionals advised me so many times . . .well, I wonder if he showed me where I could find that. . ..it was very challenging*" *(Female Unemployed) Aged 38*

"*I sometimes feel how I am going to cope with things and how I can proceed living like this. . .I have been diagnosed with mental disturbances. I am worried about the foods around me and then how I am going to take my drug.*" *Female unemployed Aged 40*

"*What can I say about my purchasing power? You might not have the chance to see what is important to you. You can't pay for what you want to eat rather, you can only pay for what you can only eat.*" *(Female self-employed Aged 35*

**Theme V: Knowledge of nutrition.**    Some participants seem to have good knowledge about nutrition and had a prior understanding of food and related issues. Also, there were participants who shared that they had poor knowledge and experience of what is better for their nutrition. Participants explained the difficult situation of comprehending what they needed for meals. Their weight loss was attributed to their previous nutritional knowledge and understanding, and they revealed that they had poor nutrition education experience at the health centre. All participants described the need for nutrition education for HIV patients, referring that many people living with HIV have a misunderstanding about managing their nutrition.

"*The physician told me that I have to take my drugs . . .but regarding what to eat I don't know. . .If he cannot advise me what to do with my eating, then it will be tough to me to understand because I am a daily labourer and I don't know the food details*" *Male Daily, labourer Aged 45*

"*I don't think they advise others too. My experience regarding this is inferior. I mean, I never heard about what to eat from this health centre. So, do you expect me to know and explain something about food and how to eat? (Laugh). . .No*" *Female Waitress Aged 30*

On the contrary, some participants struggle with accepting the need for food habit change when they are advised by health professionals. The struggling participants experienced negative motivation as it crumpled their reputation for enduring change. Participants reported, despite the fact that they believe health professionals educate them about recommended dietary changes; they had not received and even heard about the specific food types that have specific nutritional contents.

"*No one will advise not to take any specific food items. There is nothing like that. They only ask your condition. . . "female unemployed Aged 55*

**Theme VI: Support.** Participants conveyed their lack of support from parents and the community. They identified that support in the home could facilitate their lifestyle change, including assistance with meal preparation. Meanwhile, they also stated that they had inadequate appliances in the home to support meal preparation. While some felt it would be a challenge to be isolated and a meal prepared separately for them, others described ways in which following special food preparation would be helpful. Participants reported sharing experiences between patients as it could help patients who are newly diagnosed. They also reported this experience sharing has never happened throughout their experience.

"*You need to have support from your family. I can't deny this. There is nobody even at my home. I used to live with my little sister. She got married and left me alone. It was difficult. She has never seen me ever since her marriage. Because the social stigma is not decreased yet. I am a victim of that.*" Female self-employed Aged 35

" *I believe it is important, but you can't have such kind of chances. Maybe I remember a guy whom I met him in the waiting area, and it was complicated to start the conversation because they might not be interested in the discussion, and nobody has helped me to have that culture.*" Male driver Aged 55

Regarding their interaction with health professionals, most study participants were happy about the way health professionals received and treated them except for nutritional and health education at the health centers. They also stated their gender preference for females among health professionals. The respondents described their support for the present study because it provided them with hope for survival and overcoming the disease's situation and stage. Some lately diagnosed individuals discovered it was challenging to take the first step in HIV treatment. Adhering to the treatment and managing nutrition, one should get support from family, friends, society, and other HIV positive individuals who lived with the same condition. People could educate themselves by sharing experiences with other HIV positive patients. If they do not have support from others, it will not be possible for them to adhere to nutrition and drug treatment.

"*My nutritional status is becoming strengthened, and I am who I am because I got support from the health professionals, especially female health providers.. I sometimes feel as if I am at home, and they care about beyond what we eat and the drugs we take. They ask about our personal life and advice, even in the relationship we have. Can you see that it is incredible, but males don't talk that much they only ask you about the follow up only*" Male Driver Aged 40

## Discussion

This analysis identified six significant themes linked with the challenges of nutrition management among ART patients. Major themes had a substantial impact on ART patient's nutrition

and changes in their routine life to meet the expectations. All of the identified factors were inter-related (Fig 1). The identified themes were: acceptance of disease condition and health status (theme one); facilitators and barriers of treatment adherence (theme two); behavioural changes in the eating pattern (theme three); experiences of food insecurity (theme four) knowledge of nutrition (theme five); and support (theme six). Moreover, the dominant theme, which is experiences of food insecurity issues, is interlinked and bidirectional. It is, therefore, vital to controlling food insecurity to help achieve the other challenges of nutrition management.

## Acceptance of disease condition and health status

The present result on acceptance of disease condition and health status is consistent with another study. A study also indicates that most HIV positive individuals have a challenge on accepting their condition at the very beginning. It also shows that sometimes that will take a long to accept the situation when patients have engaged in risky sexual behaviour, such as inconsistent condom use, that promotes the spread of HIV [12, 13]. Specifically, participants who perceived themselves at low risk were less likely to seek testing, regardless of risky reported sexual behaviour [14].

The present study shows that people were tested with reasons that made them get tested, which led them to experience challenges in accepting their Health Status. The forceful reasons that they commonly stated were known HIV illness and death of spouse, pregnancy and leaving the country. The current study is consistent with other studies [15]. The three significant factors that have been stated as inhibiting for Voluntary Counselling & Testing (VCT) service were fear of stigma and discrimination, fear of coping with positive HIV test result, and high HIV risk perception [16]. This finding is consistent with other studies that revealed that not having a plan to disclose the status of test results negatively affected the acceptance of provider-initiated HIV testing in pregnant women [17]. Moreover, dilemmas about going to the health centers were reported by educated participants as it leads them to poor linkage to the care. It is coherent with the findings of the study conducted by access to HIV counselling and testing in Ethiopia [18].

In the current study, religion is a factor for accepting a patient's health and disease condition. The factor can be seen in two sections. According to other studies [19], religion and spirituality are the basic element for Ethiopians. They stated that spirituality has helped them cope with their problem because they will accept situations easily. This finding is consistent with another study [18] which showed that spirituality/religion helps people to cope with stressors, especially stigma/discrimination. On the contrary, the current study also shows that religion is a reason for not being able to start the medication early; rather, people with HIV prefer to go to the holy water and spend their time without starting any medication and eating well. Thus, they are prone to complicated situations and back to the health centre with serious illness. This contradiction might be due to the setting of the study area and participants' social culture [20].

## Experiences of food insecurity issues

The current study revealed that HIV patients who are on ART are suffering from different kinds of food insecurity issues. The factors are interrelated with availability, affordability, and use of foods which they consume. The food insecurity issue seemed to be a major issue for all themes such that patients felt comfortable when they are secured to manage the food availability, affordability and use. Therefore, when designing for active life expectations and managing the drugs, it is essential to consider the awareness of individuals as they do not know how to survive and live with their living standards. They felt at risk of not being able to survive. These findings are supported by different studies that revealed malnutrition among ART patients

[21–23]. Therefore, there should be nutrition care at the primary health centre, which may be a more appropriate location for all individuals to manage their nutritional status and what it necessitates.

### Facilitators and barriers of treatment adherence

The present study revealed that participants have suffered a lot of adverse reactions of medications. This also supported and coherent with a study conducted in Gondar that revealed that patients had a probability of facing three adverse effects from their ART regimen which is in contrast with a study done in Uganda, which reported that an average of five side effects could be possible; either way, the two findings from a related study setting revealed that ART drugs have side effects depending on its regimen [24]. In the current study, patients have reported repeatedly vomiting and gastrointestinal discomfort in contrast to these studies. This could probably be due to differences in ' 'patient's reports of adverse effects and the study settings, which is at the tertiary level where patients come in the more severe stages of the disease.

In this study, the critical ideas that summarize the theme were treatment knowledge and treatment know-how, emotions and passive tolerance of side effects, and nutritional advantages. Treatment knows how is low in the current study, which is inconsistent with a study conducted in Arsi [25]. Those who accessed the health institution far away (>20 km) from their home were found to more likely to use non-adherence as a coping strategy than those who were nearby (≤20 km), and this variation is might be due to the participants to the current study are using the nearby health centers because the setting is in Addis Ababa and also the study method [26].

### Behavioral changes in the eating pattern

Respondents had a change in their eating pattern ever since they had been told about their health status. The lack of clarity about the composition of foods and what to eat leads them to have a variety of changes to their lives. These findings are consistent with many studies that explore how the changes have been made according to participants; their change influences their cultural eating practice due to the attention for their eating behaviour change. Losing one's cultural practice might result from adopting a new culture. This has been seen in other studies where they assessed and have stated that to change negative health behaviours; one must first identify and promote positive health behaviours within the cultural logic of its contexts [27]. HIV patients in this study are avoiding foods that they have been told cause infections as they are immune-compromised, such as uncooked and cold foods, as it causes infections and leads to opportunistic disease [28].

On the other hand, supplemental foods are thought to be a new eating behavioural change in the current study that they are taking from health centres due to their under nutrition status. They have benefited and survived using the supplemental. This finding is consistent with the need for and use of supplemental foods for patients taking ART [29, 30].

### Knowledge of nutrition

Nutrition knowledge is found to be the basis for all aspects of the current study as nutrition is essential for people with HIV, and relevant to immunity and improving one's living condition. The more people are spending their time on a proper diet, the more they are a better at fighting their disease. According to the current study, participants do not have the basic nutrition concepts and sought information from health professionals. This is in line with other study findings such as a study conducted in Felege Hiwot referral Hospital [8] which revealed that dietary counselling was a significant factor for nutritional knowledge and sought to have

nutrition education and counselling should be given by health care workers for patients using ART to improve their nutritional knowledge. A study conducted in Addis Ababa revealed that factors related to low adherence were a low level of education; poor knowledge is helpful to adhere to food and nutrition programs [31, 32].

## Support

Regarding support and experience sharing among ART patients, in this study, patients did not have the chance to discuss issues with other patients. They highly recommend the intervention to help lead to practical changes. Some studies are consistent on how the importance of conducting this kind of program, which revealed that the difficulty of disclosure is not easy, and it is even very challenging for a couple to disclose and discuss their HIV status. However, some studies contradict the experience about sharing issues as it exacerbates social discrimination for people living with HIV [33, 34].

## Strength and limitation of the study

The study provided insight into the participant's lived experience with managing nutrition. Different factors influenced the nutrition management of patients, which was found by triangulating data with supporting methods. Thus, data and method triangulation can be the strength of this study as it is evidenced by using IDI with other promoting techniques. In this case, the current study credibility has been tested, and the study has several important strengths. The other major strength is that qualitative studies are well suited to identifying challenges from the patients' perspective. The use of an in-depth interview (IDI) approach permitted the discovery a diversity of ideas and practices which most likely would not have been detected using a quantitative approach.

The limitation of the current study is entirely shared with qualitative research, which is mainly for IPA study. By default, the researcher in this study has the privilege to construe and report what interviewees predestined within the researcher's theoretical schemes. Using this method in the current study provides room for the concept that the 'people's experiences integrate the 'researcher's views. IPA also requires theme development by the researcher by repeated reading of the actual data. Besides, all the necessary steps have been secured throughout to confirm the trustworthiness of the findings.

## Conclusion

This research found that patients with HIV were challenged with many factors around managing their nutrition and beyond taking medications. Our study identified six themes: acceptance of the disease and the health status; facilitators and barriers of treatment adherence; eating behavioral change; the experience of food insecurity issues; nutrition knowledge; and support. The controlling mechanism for nutrition management should be given the focus and aimed at a scheme line to address food insecurity. Also, there should be a practice-based teaching strategy in which HIV patients can demonstrate healthy eating habits.

## Supporting information

**S1 Checklist. COREQ checklist.**
(PDF)

**S1 File. English questionnaire.**
(PDF)

**S2 File. Amharic questionnaire.**
(PDF)

# Acknowledgments

I would like to acknowledge Wolaita Sodo University, Ethiopia, for the opportunity they created for this study. I would like to extend my gratitude for the Addis Ababa health office and the respective health centers for all aspects of direct support when I required. My acknowledgement also goes to data study participants and ART clinic coordinators in the individual health centers.

# Author Contributions

**Conceptualization:** Helen Ali Ewune.

**Data curation:** Helen Ali Ewune, Mengistu Meskele.

**Formal analysis:** Helen Ali Ewune, Mengistu Meskele.

**Funding acquisition:** Helen Ali Ewune.

**Investigation:** Helen Ali Ewune.

**Methodology:** Helen Ali Ewune, Kassa Daka, Befekadu Bekele, Mengistu Meskele.

**Project administration:** Helen Ali Ewune.

**Resources:** Helen Ali Ewune, Befekadu Bekele.

**Software:** Helen Ali Ewune.

**Supervision:** Kassa Daka, Befekadu Bekele, Mengistu Meskele.

**Validation:** Kassa Daka, Befekadu Bekele, Mengistu Meskele.

**Visualization:** Kassa Daka, Befekadu Bekele, Mengistu Meskele.

**Writing – original draft:** Helen Ali Ewune.

**Writing – review & editing:** Kassa Daka, Befekadu Bekele, Mengistu Meskele.

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
