## [Decision Letter · Decision Letter 0]

13 May 2020

PONE-D-19-35850

Challenges to nutrition management among patients on antiretroviral therapy in primary health centers’ in Addis Ababa, Ethiopia; A phenomenology study

PLOS ONE

Dear Ms Ali,

Thank you for submitting your manuscript to PLOS ONE. After careful consideration, we feel that it has merit but does not fully meet PLOS ONE’s publication criteria as it currently stands. Therefore, we invite you to submit a revised version of the manuscript that addresses the points raised during the review process.

The paper is interesting and makes a contribution to the field; however, in its current form is not ready for publication. There are a number of grammatical errors, and the manuscript would benefit from being reviewed by a language editor. I would suggest examining published PLOS One papers for the appropriate format and ensuring that your article complies with the author guidelines. For example, the Methods section in the abstract should contain details of the setting and study participants. Lastly, the title should reflect the objective of the study and should include that results are from the patient perspective. 

We would appreciate receiving your revised manuscript by Jun 27 2020 11:59PM. To enhance the reproducibility of your results, we recommend that if applicable you deposit your laboratory protocols in protocols.io, where a protocol can be assigned its own identifier (DOI) such that it can be cited independently in the future. For instructions see: http://journals.plos.org/plosone/s/submission-guidelines#loc-laboratory-protocols

We look forward to receiving your revised manuscript.

Kind regards,

Denise Evans, PhD

Academic Editor

PLOS ONE

Journal Requirements:

Additional Editor Comments (if provided):

Reviewers' comments:

Reviewer's Responses to Questions

**Comments to the Author**

1. Is the manuscript technically sound, and do the data support the conclusions?

Reviewer #1: Yes

Reviewer #2: Partly

2. Has the statistical analysis been performed appropriately and rigorously? 

Reviewer #1: N/A

Reviewer #2: N/A

3. Have the authors made all data underlying the findings in their manuscript fully available?

Reviewer #1: Yes

Reviewer #2: Yes

4. Is the manuscript presented in an intelligible fashion and written in standard English?

Reviewer #1: No

Reviewer #2: No

5. Review Comments to the Author

Reviewer #1: I have uploaded my detailed comments in the form of an attachment.

The main revisions include: 1) Proof reading and fixing multiple typos, punctuation problems, and grammatical errors; 2) Elaborate on the what nutritional management mean; 3) Rewrite the introduction; 4)Rephrase the methodology to make it succinct and improve clarity particularly regarding the description of the study area, recruitment of participants and the criteria; and 5) Rewrite the findings in relation to your research question.

Reviewer #2: In general, I find that the research aim and the results presented are important to help to improve the current situations related to nutritional interventions among HIV/AIDS patients. However, the paper has the following major gaps that should be critically addressed to be accepted:

1.Intense grammatical and typology errors, almost throughout the document. To mention few;

***Incomplete sentences: Line #46......in which other themes can be controlled with.....,auther(line #135)....reecordind(line#137). Unclear sentence: The total number of health centers in Addis Ababa is 119 and the average

101 allocation of Health centers in eight health centers per sub-city according to the population

102 coverage...(Line #100-102)

2. The introduction section lacks coherence and didn't clearly define the research problem. It has also grammatical errors. Thus, As the main aim of research is nutrition management in HIV/AIDS patients, better if the introduction begin by defining what nutrition management mean, which I didn't find the correct definition throughout the document. I would like the successive paragraphs to redefine, available evidences towards the challenges of nutrition management and the impact of nutrition non-management in patients. The final paragraph, should clearly indicate why this research is intended and what research gap it fills.

3. The study setting should be Clearly described....use the most recent population estimate (Line #99)

4. Use consistent citation style (Line #103)

5. Who are the research study participants (patients only, or, both patients and health care providers)? (Line # 110-112). It should be consistent and clearly defined

6. Participants inclusion criteria is not very clear. I am really surprised that why the research included only those speaks Amharic? (Line #115) How do you see this from the ethical point of view?. I also find that the exclusion criteria is incomplete. Do you include patients with other severe illnesses?

6. The data collection section should be better focused on and describe the data collection questionnaire, data collectors and the data collection fieldwork appropriately. As to me this section was not adequately described (Line #125).

7. Better if you describe data analysis section about what you did, rather than explaining the different steps of data analysis:-Transcription to original language, -Translation to English, -Forming codes and categories, -Forming themes etc...(Line #152)

8. Scientific rigor and quality assurance should present only what you did. (Line #169)

9. Do you secured oral or written consent? (Line 194)

10. Recommend result section to be presented in the headings below:

**Participant characteristics

** Challenges to nutrition management

***Theme 1:

***Theme 2:

***Theme 3:

***Theme 4:

***Theme 5:

***Theme 6

In addition, don't interpret findings in the result section and better if the individual responses cited with at least three characteristics: Sex, age, occupation. Interpreted findings: Line # 253-263, 352-356

11. Finally, discussion should be related to the findings presented.

6. PLOS authors have the option to publish the peer review history of their article (what does this mean?). If published, this will include your full peer review and any attached files.

Reviewer #1: Yes: Fisaha Tesfay

Reviewer #2: No

---

## [Author Response · Author response to Decision Letter 0]

18 Aug 2020

Responses to Reviewers and Editors 

We thank the editors and the reviewers. We have included our line by line response under each questions and marked the response blue colored. 

PONE-D-19-35850

Challenges to nutrition management among patients on antiretroviral therapy in primary health centers’ in Addis Ababa, Ethiopia; A phenomenology study

PLOS ONE

1.Please ensure that your manuscript meets PLOS ONE's style requirements, including those for file naming. The PLOS ONE style templates can be found at https://journals.plos.org/plosone/s/file?id=wjVg/PLOSOne_formatting_sample_main_body.pdf and https://journals.plos.org/plosone/s/file?id=ba62/PLOSOne_formatting_sample_title_authors_affiliations.pdf

We thank the editors and Reviewers for these important queries. We have corrected the file naming in which we followed style template. 

We thank the reviewers. We have included study's minimal data set as the underlying data used

This research was done on sensitive issues (ART clients) and it is not ethically permitted for us to deposit the data to the public. However, the transcripts we used and analyzed during the current study are in the main manuscript and all other necessary data sets are available from the corresponding authors on reasonable request. 

3. Additional Editor Comments (if provided):

Reviewers' comments:

Reviewer's Responses to Questions

Comments to the Author

1. Is the manuscript technically sound, and do the data support the conclusions?

Reviewer #1: Yes

Reviewer #2: Partly

2. Has the statistical analysis been performed appropriately and rigorously?

Reviewer #1: N/A

Reviewer #2: N/A

3. Have the authors made all data underlying the findings in their manuscript fully available?

Reviewer #1: Yes

Reviewer #2: Yes

4. Is the manuscript presented in an intelligible fashion and written in standard English?

Reviewer #1: No

Reviewer #2: No

5. Review Comments to the Author

Reviewer #1: I have uploaded my detailed comments in the form of an attachment.

The main revisions include:

1) Proof reading and fixing multiple typos, punctuation problems, and grammatical errors; 

The language has been edited by good language speaker and attached its evidence in track change copy. 

2) Elaborate on the what nutritional management mean; 

We have added it in the operational definition 

3) Rewrite the introduction; 

We have made corrections to make the introduction coherent 

4) Rephrase the methodology to make it succinct and improve clarity particularly regarding the description of the study area, recruitment of participants and the criteria; and 

We have made corrections depending on your attached comments 

5) Rewrite the findings in relation to your research question.

We have taken out some interpretations according to the comment given 

Reviewer #2: In general, I find that the research aim and the results presented are important to help to improve the current situations related to nutritional interventions among HIV/AIDS patients. However, the paper has the following major gaps that should be critically addressed to be accepted:

1.Intense grammatical and typology errors, almost throughout the document. To mention few;

***Incomplete sentences: Line #46......in which other themes can be controlled with.....,auther(line #135)....reecordind(line#137). Unclear sentence: The total number of health centers in Addis Ababa is 119 and the average

101 allocation of Health centers in eight health centers per sub-city according to the population

102 coverage...(Line #100-102)

Thank you so much, We have made correction according to your comments 

2. The introduction section lacks coherence and didn't clearly define the research problem. It has also grammatical errors. Thus, As the main aim of research is nutrition management in HIV/AIDS patients, better if the introduction begin by defining what nutrition management mean, which I didn't find the correct definition throughout the document. I would like the successive paragraphs to redefine, available evidences towards the challenges of nutrition management and the impact of nutrition non-management in patients. The final paragraph, should clearly indicate why this research is intended and what research gap it fills.

We have added corrections, thanks 

3. The study setting should be Clearly described....use the most recent population estimate (Line #99)

We have corrected it using recent reference, thanks

4. Use consistent citation style (Line #103)

We have cited a reference, thanks 

5. Who are the research study participants (patients only, or, both patients and health care providers)? (Line # 110-112). It should be consistent and clearly defined

Thank you, we have edited all your comments 

6. Participants inclusion criteria is not very clear. I am really surprised that why the research included only those speaks Amharic? (Line #115) How do you see this from the ethical point of view?. I also find that the exclusion criteria is incomplete. Do you include patients with other severe illnesses?

We thank you. We have corrected the exclusion and inclusion criteria as per the comment given. We have included all language speakers however the current study participants were Amaharic speakers 

6. The data collection section should be better focused on and describe the data collection questionnaire, data collectors and the data collection fieldwork appropriately. As to me this section was not adequately described (Line #125).

We have rephrased and rewrite steps in data collection 

7. Better if you describe data analysis section about what you did, rather than explaining the different steps of data analysis:-Transcription to original language, -Translation to English, -Forming codes and categories, -Forming themes etc...(Line #152)

Thank you, We made corrections depending on the comments given 

8. Scientific rigor and quality assurance should present only what you did. (Line #169)

We made corrections on the scientific rigor and quality assurance section 

9. Do you secured oral or written consent? (Line 194)

Thank you, We have secured oral and written consent 

10. Recommend result section to be presented in the headings below:

**Participant characteristics

** Challenges to nutrition management

***Theme 1:

***Theme 2:

***Theme 3:

***Theme 4:

***Theme 5:

***Theme 6

In addition, don't interpret findings in the result section and better if the individual responses cited with at least three characteristics: Sex, age, occupation. Interpreted findings: Line # 253-263, 352-356

We have corrected based on the comment given, thank you 

11. Finally, discussion should be related to the findings presented. 

We have corrected unrelated explanations with the finding 

6. PLOS authors have the option to publish the peer review history of their article (what does this mean?). If published, this will include your full peer review and any attached files.

Do you want your identity to be public for this peer review? For information about this choice, including consent withdrawal, please see our Privacy Policy.

Reviewer #1: Yes: Fisaha Tesfay

Reviewer #2: No

---

## [Decision Letter · Decision Letter 1]

9 Dec 2020

PONE-D-19-35850R1

Challenges to nutrition management among patients on antiretroviral therapy in primary health centers’ in Addis Ababa, Ethiopia; A phenomenology study

PLOS ONE

Dear Dr. Ali,

Thank you for submitting your manuscript to PLOS ONE. After careful consideration, we feel that it has merit but does not fully meet PLOS ONE’s publication criteria as it currently stands. Therefore, we invite you to submit a revised version of the manuscript that addresses the points raised during the review process.

Please have your paper thoroughly checked for grammar, spelling, and typographical errors. Failure to do so will result in further delays in processing your manuscript.  

We look forward to receiving your revised manuscript.

Kind regards,

Denise Evans, PhD

Academic Editor

PLOS ONE

Reviewers' comments:

Reviewer's Responses to Questions

**Comments to the Author**

1. If the authors have adequately addressed your comments raised in a previous round of review and you feel that this manuscript is now acceptable for publication, you may indicate that here to bypass the “Comments to the Author” section, enter your conflict of interest statement in the “Confidential to Editor” section, and submit your "Accept" recommendation.

Reviewer #2: All comments have been addressed

Reviewer #3: (No Response)

2. Is the manuscript technically sound, and do the data support the conclusions?

Reviewer #2: (No Response)

Reviewer #3: Partly

3. Has the statistical analysis been performed appropriately and rigorously? 

Reviewer #2: (No Response)

Reviewer #3: N/A

4. Have the authors made all data underlying the findings in their manuscript fully available?

Reviewer #2: (No Response)

Reviewer #3: Yes

5. Is the manuscript presented in an intelligible fashion and written in standard English?

Reviewer #2: (No Response)

Reviewer #3: No

6. Review Comments to the Author

Reviewer #2: (No Response)

Reviewer #3: 1.There is a need to be clear, correct and unambiguous (in the English used) and the manuscript presented here does not provide a concise research problem.

2. An overview of the semi-structured interview guide and a detailed selection criteria for the sample informed by literature and then snowballing in order to trace additional participants would have been useful in the methods section.

A composite summary of themes could have been a useful addition in the results section.

7. PLOS authors have the option to publish the peer review history of their article (what does this mean?). If published, this will include your full peer review and any attached files.

Reviewer #2: **Yes: **Dawit Wolde Daka

Reviewer #3: **Yes: **Nozipho O. Musakwa

---

## [Author Response · Author response to Decision Letter 1]

20 Jan 2021

Manuscript title: Challenges to nutrition management among patients using antiretroviral therapy in primary health ‘centres’ in Addis Ababa, Ethiopia: A phenomenological study

ID: PONE-D-19-35850R1 

Date: 18 January 2021 

Q. Language edition requested by the editor 

We thank the editor for this vital query. The language of our manuscript is now edited by a native English language speaker, professor Roger Watson, who is helping researchers in the author aid service platform. He has now extensively revised the English twice, and we have corrected it. The modified files (edited twice) are indicated in the manuscript with track changes 1 and 2.

---

## [Decision Letter · Decision Letter 2]

3 Mar 2021

PONE-D-19-35850R2

Challenges to nutrition management among patients on antiretroviral therapy in primary health centers’ in Addis Ababa, Ethiopia; A phenomenology study

PLOS ONE

Dear Dr. Ali,

Thank you for submitting your manuscript to PLOS ONE. After careful consideration, we feel that it has merit but does not fully meet PLOS ONE’s publication criteria as it currently stands. Therefore, we invite you to submit a revised version of the manuscript that addresses the points raised during the review process.

Please consider the Reviewers advice to check the manuscript for language and grammatical errors to improve the quality of your manuscript. 

We look forward to receiving your revised manuscript.

Kind regards,

Denise Evans, PhD

Academic Editor

PLOS ONE

Journal Requirements:

Reviewers' comments:

Reviewer's Responses to Questions

**Comments to the Author**

1. If the authors have adequately addressed your comments raised in a previous round of review and you feel that this manuscript is now acceptable for publication, you may indicate that here to bypass the “Comments to the Author” section, enter your conflict of interest statement in the “Confidential to Editor” section, and submit your "Accept" recommendation.

Reviewer #3: All comments have been addressed

2. Is the manuscript technically sound, and do the data support the conclusions?

Reviewer #3: Yes

3. Has the statistical analysis been performed appropriately and rigorously? 

Reviewer #3: Yes

4. Have the authors made all data underlying the findings in their manuscript fully available?

Reviewer #3: Yes

5. Is the manuscript presented in an intelligible fashion and written in standard English?

Reviewer #3: No

6. Review Comments to the Author

Reviewer #3: The English needs to be tightened up more to ensure that it is clear, correct and unambiguous. Overall an important study to find out from patients what stops them from adhering to treatment from a nutritional point of view

7. PLOS authors have the option to publish the peer review history of their article (what does this mean?). If published, this will include your full peer review and any attached files.

Reviewer #3: **Yes: **Nozipho O. Musakwa

---

## [Author Response · Author response to Decision Letter 2]

23 Mar 2021

Thank you very much! It was very helpful. It was a great opportunity. I would like to thank the editors and reviewers for their fruitful support.

---

## [Editor Report · Decision Letter 3]

19 Apr 2021

Challenges to nutrition management among patients using antiretroviral therapy in primary health ‘centres’ in Addis Ababa, Ethiopia: A phenomenological study

PONE-D-19-35850R3

Dear Dr. Ali,

We’re pleased to inform you that your manuscript has been judged scientifically suitable for publication and will be formally accepted for publication once it meets all outstanding technical requirements.

Kind regards,

Denise Evans, PhD

Academic Editor

PLOS ONE
---

## [Editor Report · Acceptance letter]

2 Jun 2021

PONE-D-19-35850R3 

Challenges to nutrition management among patients using antiretroviral therapy in primary health ‘centres’ in Addis Ababa, Ethiopia: A phenomenological study 

Dear Dr. Ali Ewune:

I'm pleased to inform you that your manuscript has been deemed suitable for publication in PLOS ONE. Congratulations! Your manuscript is now with our production department. 

Kind regards, 

on behalf of

Dr. Denise Evans 

Academic Editor

PLOS ONE